# Advances in the Management, Treatment, and Surveillance of Anal Squamous Cell Cancer

**DOI:** 10.3390/cancers17081289

**Published:** 2025-04-10

**Authors:** Cynthia Araradian, Maura Walsh, Hayley Standage, Vassiliki Liana Tsikitis

**Affiliations:** Department of Surgery, Oregon Health & Science University, Portland, OR 97239, USA; walshma@ohsu.edu (M.W.); standagh@ohsu.edu (H.S.); tsikitis@ohsu.edu (V.L.T.)

**Keywords:** anal cancer, HSIL, Nigro protocol, immunotherapy, squamous cell cancer

## Abstract

Anal cancer is rare compared to other gastrointestinal cancers, but its incidence has risen over the past decade. Advances in cancer screening and detection, as well as the identification of the cancer’s precursor, squamous intraepithelial lesions, have led to more effective treatment and outcomes of the disease. Key clinical trials have shaped the current approach to anal cancer treatment. The Nigro protocol, which includes simultaneous chemotherapy and radiation, is the cornerstone of anal cancer treatment. We review several important studies that have identified the most effective chemotherapy and radiation regimens and explore the addition of immunotherapy to cancer treatment. These advances in the management of anal cancer have contributed to improved outcomes for patients. Future research will be needed to further explore the role of biomarkers, treatment with immunotherapy agents, and personalized treatment plans.

## 1. Introduction

### 1.1. Anal Cancer

Anal cancer cases have been rising by, on average, 2.2% each year over the past decade, accounting for 0.5% of all new cancer cases in 2024 [1]. Anal cancers are classified by their anatomical location and their cells of origin. For this reason, it is important to understand the anatomy of the anus and rectum. The anus is distal to the rectum and connects to the outside of the body. The surgical anal canal measures 2–4 cm long and is the area from the anorectal ring containing the anal sphincter complex and puborectalis muscle. The anatomic anal canal extends from the dentate line to the anal verge, which denotes the boundary between the hairless anoderm and the perianal skin. The anatomy of the dentate line is critical. Proximal to the dentate line, the epithelium is a combination of squamous, columnar, and transitional epithelium. The skin from the dentate line distally to the anal verge is a modified squamous epithelium. The anal margin extends radially from the anal verge and contains the surrounding skin. Often, cancers of the anal margin are treated as skin cancers.

The types of anal cancer by cell origin consist of squamous cell carcinoma, which is the most common variant, melanoma, neuroendocrine, adenocarcinoma, basal cell carcinoma, lymphoma, and extramammary Paget’s disease. Given the histology of the different areas of the anus, usually below the dentate line the malignancy is squamous cell carcinoma of anus (SCCA). Given the varying histology above the dentate line, this can include different variants of squamous cell cancer (i.e., cloacogenic, mucoepidermoid, basaloid, or transitional). Regardless, they all have the same work-up and treatment [2].

### 1.2. Diagnosing and Staging Anal Cancer 

Patients can present asymptomatic or with symptoms concerning for a malignant process. The most common presenting symptoms are rectal bleeding, pain, nonhealing ulcers, nonhealing fistulas, or a mass. The initial exam should consist of a digital rectal exam (DRE), anoscopy, and inguinal lymph node palpation [3]. Anal cancers are diagnosed with tissue biopsies of suspicious lesions or masses. Once a diagnosis of SCCA is made, staging workup consists of a CT chest, abdomen, and pelvis. Pelvic MRI is obtained to assess the location of the mass in relation to the anal canal or anal margin. FDG-PET scans can be used to assess lymph node involvement and distant metastasis with a higher sensitivity for abnormal lymph nodes [4]. If concerning lymph nodes are identified on imaging or palpated on exam, fine needle aspiration of the involved lymph nodes is indicated.

The American Joint Committee on Cancer system for staging is the most commonly used criteria (Table 1) [5].

Table 1 does not consider the category of microinvasive T1 anal cancers considered superficially invasive squamous cell carcinoma (SISCCA). The implementation of a definition for SISCCA was to identify patients who could be managed with local excision as opposed to undergoing systemic treatment. The consensus definition for SISCCA of the anal canal is a carcinoma with an invasive depth ≤3 mm from the basement membrane, horizontal spread of ≤7 mm in maximal extent, and has been completely excised [6]. Currently, besides SISCCA, most anal cancers are treated with a modified Nigro protocol. 

### 1.3. Nigro Protocol and Current NCCN Guidelines 

Since 1974, the Nigro protocol has been first line treatment for SCCA of the anal canal [7]. The current National Comprehensive Cancer Network (NCCN) guidelines include the treatment of early and locally advanced anal cancer with chemoradiation (Figure 1) [8].

Although the mainstay treatment for early and locally advanced anal cancer is chemoradiation, there are also surgical options that relate to the primary treatment or salvage therapy. In regard to primary treatment of anal cancer there is uncertainty about the use of chemoradiation for T1-2, node negative SCCA given this may be overtreating the disease. There are ongoing clinical trials to assess the efficacy of local excision. Currently, the NCCN guidelines suggest the use of local excision for SISCCA and for perianal cancers. For the perianal lesions, NCCN recommends excision with 1 cm margins [8]. At this time, NCCN guidelines do not mention local excision for anal canal lesions. A study by Arana et al. performed local excision with >2 mm margins for anal canal lesions and >1 cm margins for perianal lesions with subsequent chemoradiation for patients who had inadequate margins [9]. Given the current lack of high-quality data for local excision, the ongoing trials, like ACT 3, will provide data to help demonstrate the clinical utility.

Abdominoperineal resection (APR) is now considered a salvage operation for patients who fail locoregional treatment with chemoradiation. The APR leads to a permanent colostomy with the possibility of perineal wound and surgical complications. Based on a systematic review published in 2019, the median 5-year overall survival following an APR was 39%, with 24% of patients developing recurrent disease and 9% of patients developing metastatic disease after the operation was completed [10].

Currently, NCCN Guidelines recommend systemic chemotherapy with immunotherapy (pembrolizumab, nivolumab, or retifanlimab) as a second-line systemic option for metastatic disease. It can also be considered for patients who have already undergone the Nigro protocol and have persistent or recurrent disease prior to APR. However, patients may benefit from earlier treatment with immunotherapy (Figure 1).

### 1.4. Precursors to Anal Cancer

SCCA has precursor lesions known as anal intra-epithelial neoplasia (AIN) or squamous intraepithelial lesions (SILs). There has been a shift towards predominantly using SIL for nomenclature since the creation of the Lower Anogenital Squamous Terminology project, which was created to help standardize the naming of these pathologic diagnoses [6]. The precursor lesions known as low-grade anal squamous intraepithelial lesions (LSILs) correspond to AIN1, while high-grade anal squamous intraepithelial lesions (HSILs) correspond to AIN 2 and AIN 3. The precursor lesions are often compared to cervical, vulvar, and vaginal intra-epithelial neoplasia, and much of the knowledge about anal precursor lesions has been adopted from this literature. Both anal precursor lesions and genital precursor lesions are associated with human papillomavirus (HPV), most specifically the high-risk subtypes HPV 16 and 18. Most anal cancers are caused by HPV infection. Risk factors for anal cancer include individuals living with human immunodeficiency virus (HIV), men who have sex with men, immunosuppressed individuals, women with known cervical dysplasia and those with high-risk sexual practices [3]. The risk for individuals with HIV is highest; men living with HIV have a higher risk than women living with HIV [11].

The goal of this review is to discuss instrumental trials that have shaped the management of anal precursor lesions and anal cancer over the last two decades.

## 2. Clinical Trials: Screening and Treatment of HSIL

### 2.1. ANCHOR Study

The precursor lesion for anal cancer is considered to be HSIL. Much of the research about HSIL and surveillance has been modeled after cervical cancer guidelines and screening for precursor lesions. The American Society of Colon and Rectal Surgeons (ASCRS) clinical practice guidelines recommend that patients with anal dysplasia are regularly monitored with history and physical exams along with consideration of anal Pap tests [3]. There were no randomized control trials that discussed the benefits of treatment of HSIL prior to the publication of the Anal Cancer- HSIL Outcomes Research (ANCHOR) study in 2022 [12]. The ANCHOR study is a phase 3 randomized control trial that took place at 25 study sites in the US. It included individuals living with HIV over the age of 35 with a diagnosis of HSIL on high-resolution anoscopy (HRA). The patients were randomized into treatment of HSIL or the active monitoring group. The treatment group was treated with ablative procedures or topical therapies. These patients were followed with HRAs completed every 6 months. There was an overall 57% lower risk of anal cancer in the treated group compared to the active monitoring group over a median follow up of about 2 years [12]. This study highlights the importance of providing access to clinics with HRA-trained physicians for patients living with HIV. It also set the stage for earlier detection of anal cancers with routine, active screening of high-risk patients.

### 2.2. Clinical Trials for Treatment of HSIL

HSIL is treated with a range of therapeutic approaches including topical (imiquimod or fluorouracil) or procedural (electrocoagulation or infrared coagulation) interventions. Clinical trials have been conducted to evaluate the outcomes between the treatment modalities. A trial completed in the Netherlands included HIV-positive men who have sex with men who were assigned treatment of their anal intra-epithelial neoplasia with 16 weeks of imiquimod (3 times per week), 16 weeks of fluorouracil (twice per week), or monthly electrocautery (for 4 months) [13]. The patients had all grades of anal intra-epithelial neoplasia. Patients were monitored with an HRA 4 weeks post treatment, then had subsequent HRAs at 24, 48, and 72 weeks. One hundred and fifty-six patients were randomly assigned to their treatment group with results indicating a complete response in 13/54 (24%) patients treated with imiquimod, 8/48 (17%) patients in the fluorouracil group, and 18/46 (39%) patients in the electrocautery group. Overall, the study findings indicated that electrocautery was associated with an improved complete response rate, and patients undergoing this treatment reported fewer and less severe side effects [13].

Another study completed by seven US AIDS Malignancy Consortium sites recruited HIV-positive patients with HSIL to a randomized trial comparing ablation with infrared coagulation to active monitoring [14]. One hundred and twenty-one patients were enrolled and followed for 24 months. At 1 year, patients that underwent ablation were significantly more likely to clear their HSIL lesions. Patients in the infrared coagulation group had minimal side effects with the majority of symptoms reported as pain or postoperative bleeding [14].

Overall, both studies assessing the treatment options for HSIL demonstrated the benefits for electrocautery or infrared coagulation use to treat HSIL with minimal side effects. It is also important to note that HIV-positive patients were included in both cohorts which is an important distinction for these trials compared to the trials that assess treatment of anal cancer discussed later in this review. It is crucial to include HIV-positive patients into these cohorts as these patients tend to be at highest risk, requiring surveillance and treatment.

## 3. Clinical Trials for Treatment of Anal Cancer

Advances in anal cancer treatment have been influenced by clinical trials published over the last three decades (Table 2).

### 3.1. Locoregional Disease

#### 3.1.1. ACT I

One of the first major clinical trials after the Nigro protocol’s introduction was the Anal Cancer Trial (ACT) I published in 1996. This study randomized 585 patients from multiple centers in the UK. The goal of this study was to compare radiation alone to concurrent chemoradiation with 5-FU and mitomycin. They included patients with non-metastatic anal cancer and excluded T1N0 patients as they felt these patients would benefit from local excision without the need for systemic therapy. The study’s primary end point was local failure rate, assessed 6 weeks after the initial treatment to 6 months after therapy. If patients had greater than a 50% or complete response at 6 weeks then a radiation boost was recommended, if patients had less than a 50% response then they were recommended to have an APR. They had a median follow up of 42 months. They reported a significant reduction in local failure rate in the concurrent chemoradiation group with 101/283 (39%) patients reported to have local failure compared to the radiation alone group of 164/279 (61%) patients. There was no statistically significant difference in overall survival. With the chemoradiation group, there were initially higher morbidity rates, but this appeared to balance out with the radiation only group at more than 2 months from treatment. This study was instrumental in highlighting the importance of concurrent chemoradiation in the treatment of anal cancer [15].

#### 3.1.2. RTOG 98-11

The RTOG 98-11 study was a multicenter, phase III randomized control trial that took place in centers across the United States. Patients with non-metastatic anal cancer were included with the exclusion of T1 disease and those with severe comorbid conditions, including acquired immunodeficiency syndrome (AIDS). The study’s treatment arms included (1) radiation with 5-FU and mitomycin and (2) radiation with 5-FU and cisplatin. It was interesting to note that the control group of 5-FU and mitomycin was administered with concurrent radiation and the experimental group was given 5-FU and cisplatin as an induction dose of 2 cycles followed by concurrent chemoradiation. This experimental group was created to try to sensitize the tumor with cisplatin before the radiation. The primary endpoint for this study was disease free survival. Overall, there were 644 patients included in the trial with a median follow up of 2.51 years. The initial study reported estimated 3- and 5-year disease free survival rates of 67% and 60% for the mitomycin group and a 61% and 54% rate for the cisplatin group. They reported their results earlier than anticipated, since by their second interim analysis, they had enough data to determine that there was no statistically significant difference between the groups and additional data would not alter this statistical finding. The overall survival rates were also not statistically different between the groups. The one important statistically different variable was the colostomy rates with a 10% colostomy rate at 3 and 5 years for mitomycin and 16% and 19% for 3 and 5 years, respectively, for the cisplatin group [16].

An update in 2012 demonstrated that concurrent chemoradiation with 5-FU and mitomycin did in fact have statistically significant improvement in 5-year disease free survival and overall survival compared to the induction and concurrent chemoradiation of 5-FU and cisplatin. In addition, induction chemotherapy did not seem to provide any benefit as it delayed definitive treatment [17]. The results were instrumental in demonstrating that there was no benefit to sensitization of the tumor with additional chemotherapy and may in fact cause more harm.

#### 3.1.3. ACCORD 3

The ACCORD 3 trial published in 2012 utilized a 2 × 2 factorial design to assess the impact of induction chemotherapy and increased dose of radiation. The four treatment groups were as follows: (1) induction chemotherapy followed by concurrent chemoradiation followed by a standard boost, (2) induction chemotherapy with concurrent chemoradiation followed by a high dose radiation boost, (3) concurrent chemoradiation followed by a standard boost, and (4) concurrent chemoradiation followed by a high dose boost. All the chemotherapy regimens utilized were 5-FU on day 1–4 and cisplatin on day 1. The concurrent chemoradiation was delivered as 45 Gy over 25 fractions while the boost doses varied. The standard boost was given as 15 Gy over 8 fractions while the high dose boost was given as 20 or 25 Gy over 8 fractions. The primary end point was colostomy free survival. They included non-metastatic anal cancer patients and included patients with HIV. The 307 patients included in the trial had a median follow up time of 50 months. They reported colostomy free survival of 69.6%, 82.4%, 77.1%, and 72.7% for arms 1, 2, 3, and 4, respectively. Similarly to the RTOG 98-11 trial, the results did not demonstrate an advantage to the induction chemotherapy. They also demonstrated there was no benefit to high dose boost of radiation [18]. Although this trial, contrary to others, did not exclude HIV positive patients, they did not report on the number of HIV patients included.

#### 3.1.4. ACT II

Although the ACT II trial had a similar methodology and intent to RTOG 98-11, there were few notable differences. The ACT II trial included the role of maintenance chemotherapy and surveillance of patients. Differences included patients and care teams were blinded to the treatment groups except for patients with T1 cancers and HIV-positive patients were excluded. The regimens tested included radiation with mitomycin and 5-FU or cisplatin and 5-FU. The 2 × 2 component included an addition of maintenance doses of the chemotherapy regimen to half the patients in each respective treatment arm after the completion of chemoradiation. The intent of the maintenance dose was to determine if use of two additional doses of chemotherapy would prevent metastatic disease. The study enrolled 940 patients with anal canal and anal margin non-metastatic cancers. The primary endpoints for the study included complete response at 26 weeks, acute grade 3–4 adverse events for 4 weeks after chemoradiation, and progression-free survival, specifically for the maintenance therapy. The median follow-up time was 5.1 years. There was a complete response at 26 weeks for 386/431 (89.6%) patients in the cisplatin group and 391/432 (90.5%) patients in the mitomycin group, which was not statistically significant. The proportion of patients with grade 3–4 adverse events in each group was 334/472 (71%) for the mitomycin group and 337/468 (72%) for the cisplatin group, showing no difference between the treatment regimens. In regard to the maintenance chemotherapy, only 196/448 (44%) completed the two doses of chemotherapy, while another 119/448 (27%) had dose modifications. The progression-free survival was not statistically significant between the maintenance (74%) and no maintenance (73%) groups and was similar between the cisplatin (72%) and mitomycin (73%) groups [19]. Overall, the study highlighted that there was no benefit to maintenance chemotherapy and that the standard of care should remain as 5-FU and mitomycin, especially with the decreased doses of chemotherapy needed with this regimen.

#### 3.1.5. RTOG 0529

Many clinical trials have focused on the chemotherapy component of chemoradiation, but the RTOG 0529 trial focused on radiation with a standardized chemotherapy regimen. The RTOG 0529 trial utilized dose-painted intensity modulated radiation therapy with doses of radiation for target volumes according to a patient’s stage. This radiation treatment was performed in conjunction with 5-FU and mitomycin therapy. This was a phase II trial which enrolled and evaluated 52 patients from December 2006 to March 2008. Patients with T1 or M1 disease and severe comorbid medical conditions, including AIDS, were excluded. The study did not specify if patients with well-controlled HIV were included. Fifty-one patients completed the dose painted intensity modulated radiation therapy with a median duration of therapy of 43 days. Grade 2+ GI and GU adverse events were seen in 40/52 (77%) patients, which was comparable to the RTOG 98-11 study. They reported a significant reduction in hematologic and dermatologic toxicity and GI and GU morbidity by attempting to spare the small bowel [20]. They noted the understanding and use of intensity modulated radiation therapy can represent a challenge as it requires an understanding of the planning CT with the appropriate calculations of clinical target volume. Currently, intensity modulated radiation therapy is the standard radiation therapy used for chemoradiation as it has better targeting of the area while minimizing damage to surrounding organs and tissues.

#### 3.1.6. Summary of Clinical Trials for Locoregional Disease

It is important to note that T1 anal cancers were often excluded from these trials. There is no explanation often given for their exclusion, but it is important to identify as a clinician given the increasing number of trials and studies including new pharmacologic treatments but also local excision. Given the exclusion of early-stage anal cancer patients, it is difficult to determine the ideal therapeutic regimen with concerns for overtreatment with use of chemoradiation. The exclusion of patients with HIV also presents a clinical challenge as these patients tend to be at highest risk for SCCA. Although patients with HIV are treated with chemoradiation, they may be at higher risk for toxicity, so their inclusion into trials would be critical to assess these risks.

The above clinical trials have been critical in establishing the standard of care for SCCA treatment. ACT I demonstrated the benefit for chemoradiation over radiation. The RTOG 98-11 trial helped established the superiority of mitomycin and 5-FU over cisplatin-based regimens. RTOG 98-11 and ACCORD 3 trials failed to demonstrate a benefit to induction chemotherapy which led to delays in primary treatment. ACT II interestingly tested maintenance chemotherapy, again showing no benefit. Lastly, RTOG 0529 helped establish intensity modulated radiation therapy as the standard of care. Given the data presented in these trials, the current standard of care for non-metastatic disease includes chemoradiation with intensity-modulated radiation therapy and concurrent 5-FU and mitomycin.

### 3.2. Metastatic Disease

Metastatic disease in SCCA occurs in about 10% of patients at initial diagnosis. The most common sites of metastatic disease include the liver, lung, extra-pelvic lymph nodes, peritoneum, and bone [21]. Currently, stage IV anal cancer has about a 22.1% overall survival at 5 years [5].

The primary treatment modality in the setting of metastatic disease includes systemic chemotherapy with second line options including immunotherapy [8]. A study utilizing the National Cancer Database evaluated the use of subsequent radiation to the primary in the setting of metastatic disease. They reported improved median 2- and 5- year overall survival in those who received chemoradiation compared to chemotherapy alone [22]. Given this study utilized a national database that does not include critical information (i.e., type of chemotherapy agent), no definitive conclusion can be made about the use of chemoradiation. The remainder of this section will focus on randomized control trials utilizing chemotherapy and the next section will focus on immunotherapy for metastatic disease.

#### InterAAct 

InterAAct was a randomized phase II clinical trial published in 2020, including 91 patients with metastatic or locally recurrent inoperable SCCA. Patients were chemotherapy naive and recruited from 60 centers. Patients with HIV were included in this study if they were on highly active antiretroviral therapy, although only five patients were HIV positive at enrollment. Patients were assigned to the carboplatin and paclitaxel group or cisplatin and 5-FU group. Patients were treated for 24 weeks or until adverse effects or disease progression. The primary end point in this study was the overall response rate with secondary end points consisting of overall survival (OS), progression-free survival (PFS), disease control, and adverse effects. They had an overall median follow up of 28.6 months. The results indicated that carboplatin and paclitaxel had an overall response rate of 59%, overall survival of 20 months, severe adverse events in 15/42 (36%) patients and 6/39 (15.4%) patients experienced disease progression in the follow-up time. For the cisplatin and 5-FU group, there was an overall response rate of 57%, overall survival of 12.3 months, severe adverse effects in 26/42 (62%) patients and 8/35 (22.9%) patients had disease progression [23]. Although survival was improved with the carboplatin-paclitaxel regimen, the patients had more adverse events that occurred. Based on the results of the InterAAct trial, the NCCN updated their guidelines to include carboplatin and paclitaxel as the first line recommendation for metastatic disease.

## 4. Clinical Trials for Immunotherapy

Advances in immunotherapy have led to its inclusion in the management of many different cancers. Its role in SCCA is still evolving but was included in the NCCN guidelines in 2018 due to a few of the trials listed below [24,25]. Currently, NCCN guidelines recommend immunotherapy as a second-line systemic therapy for metastatic disease or as an option for recurrent disease prior to proceeding with APR (Figure 1).

Tumors often highly overexpress cell programmed death-ligand 1 (PD-L1) which binds to cell surface receptor programmed death-1 (PD-1) expressed on T cells (Figure 2A). Once bound, the downstream cascade inhibits T cell activation. In a healthy individual, this is an essential pathway to prevent the immune system from attacking itself. However, in this instance, it allows the tumor to evade immune detection and anti-tumor response [26]. This immune checkpoint pathway is the target of anti-PD-1 monoclonal antibodies (i.e., pembrolizumab, retifanlimab, nivolumab) and anti-PD-L1 antibodies (i.e., avelumab), which prevent PD-L1, found on tumor cells, from binding PD-1, thereby enabling T cell activation (Figure 2B,C).

### 4.1. NCI9673

The NCI9673 study evaluated the efficacy of nivolumab for patients with previously treated unresectable metastatic anal cancer in a multicenter, phase II trial at 10 academic US centers which included 37 patients. A total of 9 (24%) patients achieved a response, 7 partial and 2 complete, and 7/9 (78%) of those patients achieved a durable response. Median progression-free survival was 4.1 months, but the longest response exceeded 1 year and the data cutoff date. Overall, 14% of the cohort experienced a grade 3 adverse event. Interestingly, on analysis of 13 tumor biopsies, there were higher rates of CD8 T cells and tumors expressing PD-L1 in the responders than the non-responders. There was a low mutation rate in this cohort; researchers proposed that immunogenicity drove the response to immunotherapy in reaction to the HPV infection rather than neoantigens due to the mutational burden. This study demonstrated that nivolumab was well tolerated in their cohort and could be a promising option for patients with refractory metastatic SCCA [25].

### 4.2. KEYNOTE 28

KEYNOTE 28 evaluated the safety and efficacy of pembrolizumab in a multicohort, phase IB trial in 24 patients with locally advanced or metastatic SCCA. They only included patients with tumors positive for PD-L1 expression. Patients were treated every 2 weeks for up to 2 years with response assessed by imaging (CT or MRI). Most of the patients, 13/24 (54%) had previously been treated with another treatment modality. Four (17%) patients had a partial response with pembrolizumab, and there was a 58% disease control rate [24].

### 4.3. KEYNOTE 158

KEYNOTE-158 trial was a multicenter, non-randomized, phase II study that investigated the efficacy of using pembrolizumab for patients with locally advanced or metastatic SCCA who had previous failure or intolerance to standard therapy. Of 112 patients, 6 (5%) had a partial response and 6 (5%) had a complete response. In comparing PD-L1 positive and PD-L1 negative tumors, 11/75 patients with PD-L1 positive tumors had an objective response compared to 1/30 in the PD-L1 negative group. Overall survival at 24 months was estimated at 26%. While treatment related adverse events occurred in 68 (61%) patients, only 12 (11%) patients experienced a serious treatment-related adverse event [28].

### 4.4. POD1UM-202

POD1UM-202 was a single-arm, multicenter, phase II study that evaluated the partial/complete response rates with retifanlimab treatment in 94 patients with previously treated advanced or metastatic SCCA. It also demonstrated an acceptable safety profile and durable antitumor activity, with an overall response rate of 13 (13.8%), with 1 complete and 12 partial responses and stable disease in 33 (35.1%) patients. Interestingly, none of the responders’ tumors demonstrated deficient DNA mismatch repair status, and responses were observed in patients who were PD-L1 negative. This study included HIV positive patients, and retifanlimab was well tolerated in these patients; there was no emergence of opportunistic infection or loss of HIV control. In an effort to identify biomarkers that may identify responders to immunotherapy, this study showed a positive correlation between tumor inflammation mRNA signature and overall survival and PD-L1 mRNA expression and overall survival [29].

The results of the randomized phase III, double blind trial of POD1UM-303, comprise the newest trial that compares the use of retifanlimab in combination with the standard of care chemotherapy compared to placebo with chemotherapy. Three hundred and eight patients were enrolled with improved progression-free survival in the arm that receivied retifanlimab [30]. The complete results have not yet been published, but it demonstrates favorable results for this treatment regimen.

### 4.5. CARACAS

The above studies have demonstrated a modest response rate using immunotherapy as a single agent, but its role in combination with other treatment modalities has yet to be defined. In the CARACAS study, a combination therapy with avelumab and cetuximab was evaluated for patients with advanced SCCA who progressed on other therapies in an open-label, multicenter, “pick the winner” randomized control trial in Italy. Avelumab is an anti-PD-L1 therapy, and cetuximab is an anti-epidermal growth factor receptor agent. The study found that partial or complete response rates were higher in those who were treated with both avelumab and cetuximab (17%) versus avelumab alone (10%) [31]. Given the promising results with combined therapy, further investigation into this treatment with minimal side effects can potentially alter the future of metastatic anal cancer treatment.

### 4.6. Other Immunotherapy

In addition to anti-PD-1 and anti-PD-L1 therapies, other targets are currently being explored, including relatlimab (anti-LAG3), daratmumab (anti-CD38), and ipilimumab (anti-cytotoxic T-lymphocyte antigen 4, CTLA-4) [32]. A phase II trial, NCI 9673 part B, created a combination immuotherapy regimen including nivolumab and ipilimumab. There was no statistically significant improvement in progression-free survival or overall survial with the addition of ipilimumab to nivolumab therapy [33]. Further investigation into combination immunotherapy compared to combination immunotherapy and chemotherapy will be crucial to determine ideal treatment regimens.

Given the strong association between HPV and SCCA, there are also currently bioengineered immunotherapy options created to induce immunogenic responses against the tumor. Axalimogene filolisbac (ADXS11-001) is a nonpathogenic, live, and attenuated strain of the bacteria Listeria monocytogenes which has been engineered to stimulate the immune system by secreting a fusion protein of Listeriolysin O and HPV-16 E7 to act as an antigen [34]. Patients were given this treatment as a single agent if they had refractory metastatic SCCA. The study had loss of patients given disease progression, withdrawal of consent, or adverse events, so the primary outcome was not achieved. This study may not have been completed, but we suspect this is one of many bioengineered therapies that will be studied for SCCA.

## 5. Surveillance

### 5.1. Surveillance Guidelines

Once the Nigro protocol became standard of care for SCCA, there was uncertainty in the optimal surveillance timeframe before assessing patient response to treatment. Guidelines initially suggested follow up around 6–12 weeks for response, and patients were subsequently recommended to undergo salvage APR and/or additional treatments if they did not have a complete response. Utilizing patient data from the ACT II trial, the authors explored the optimal amount of time to wait before deciding about a patient’s response status post-treatment. They were assessed at 11, 18, and 26 weeks from the start of chemoradiotherapy, where rates of complete clinical response were 52%, 71%, and 78%, respectively. Rates of overall survival were 85–87% within patients who had a complete clinical response, and eventual outcomes were independent of when complete clinical response was achieved [35]. This study demonstrated that a longer interval prior to assessment was optimal; NCCN guidelines now reflect this, recommending assessment with rectal exam at 8–12 weeks with re-evaluation at 4 weeks for persistent disease (Table 3). It is then recommended to observe and re-evaluate at 3-month intervals. If still persistent at 6 months, the lesion should be re-biopsied and restaged. Salvage APR or first-line systemic therapy is indicated for locally persistent/recurrent disease or metastatic disease, respectively [8].

As for patients who have undergone the Nigro protocol with complete clinical response or patients with T1 disease who have undergone local excision with adequate margins, patients undergo a surveillance protocol that involves serial exams and imaging (Table 3) [8]. For the first 5 years, patients should have a digital rectal exam and inguinal node palpation every 3–6 months. For 3 years, anoscopy should be performed every 6–12 months, as well as annual CT chest/abdomen/pelvis with contrast. Yearly abdominal MRI with contrast and CT chest without contrast is an acceptable alternative (Table 3) [8].

Local recurrence should be subsequently treated with APR. Similarly, post-operative surveillance is with the same imaging and physical exam protocol. Recurrence in the inguinal nodes can be treated with groin dissection. Those who have not yet had chemoradiation to the groin may be offered this protocol, otherwise they can undergo chemotherapy regimens. Notably, FDG-PET scan is not part of SCCA surveillance [8].

### 5.2. Biomarkers

Biomarkers used in the diagnosis and surveillance of SCCA would provide a non-invasive method of screening patients as well as monitoring treatment progression or regression. The use of biomarkers can also help create personalized therapies and aid in decision making. This section will provide a brief overview of the biomarkers currently being tested for use in SCCA.

#### 5.2.1. Circulating Tumor DNA

Circulating tumor DNA, also known as ctDNA, specifically for HPV, identifies the circulating HPV DNA in the blood of patients. Often, ctDNA is used as a method to test for residual disease in patients who have undergone treatment. A study has shown that the detection of HPV ctDNA 3 months after the completion of chemoradiation was associated with recurrence and decreased recurrence free survival [36]. Patients enrolled in the Epitopes-HPV02 trial were also monitored with HPV ctDNA at baseline and post-treatment with docetaxel, cisplatin and 5-FU. Patients with residual positive HPV ctDNA at the end of the chemotherapy had shorter progression free survival and a reduction of 1-year overall survival [37]. Circulating tumor DNA appears to be an effective method to test for recurrence or progression of disease, but given it is also used for other HPV associated cancers including cervical and head and neck, it may be difficult to use it as an initial diagnostic tool given its lack of specificity.

#### 5.2.2. Tumor-Infiltrating Lymphocytes

Tumor-infiltrating lymphocytes, known as TILs, are a part of the host’s immune response against the tumor. The degree of tumor-infiltrating lymphocytes has been well studied in multiple cancers as a prognostic marker. The tumor sample is used for testing the tumor-infiltrating lymphocytes with higher infiltration corresponding to improved relapse-free survival. Patients with high TIL scores had a recurrence free survival of 92.3% compared to those with a minimal or absent TIL who had a recurrence free survival of 78.3% [38]. The assessment of TIL in each tumor sample can be helpful especially with the addition of immunotherapy for treatment.

## 6. Future Directions

The treatment regimen of concurrent chemoradiation for locally advanced anal cancer has been enhanced with the above listed clinical trials. Currently, there is a shift towards trying to individualize treatments and de-escalate therapies for anal cancer based on the patient’s stage to minimize adverse events. The DECREASE study, a phase II clinical trial, is testing lower doses of chemoradiation for patients with early stage SCCA. The DECREASE study is utilizing decreased fractions of intensity modulated radiation therapy to 20 or 23 in comparison to standard of care of 28 fractions with concurrent mitomycin and 5-FU or capecitabine. They categorize early anal cancer as T1-T2 node negative SCCA. This study includes patients with HIV with CD4 counts greater than 300. This study is estimated to be completed in 2029 [39]. The Personalizing Radiotherapy Dose in Anal cancer is the overarching name given to the trials of ACT 3, 4, and 5. Each ACT is set to examine different components of the treatment regimen; ACT 3 is evaluating local excision for T1N0 anal margin cancers, ACT 4 is evaluating radiation dose reduction for T1-2N0 tumors with capecitabine, and ACT 5 is evaluating different radiation doses with either 5-FU or capecitabine in locally advanced cancers (T3-4N1-3) [40]. There are also ongoing clinical trials that are evaluating the use of immunotherapy as a first-line treatment modality in combination with chemotherapy for advanced anal cancer [28]. Additionally, more research must be performed to identify biomarkers to predict who will have the best response to immunotherapy.

Given the strong association between HPV and anal cancer, the utilization of the HPV vaccine may play a role in the incidence of anal cancer over the next several decades. The quadrivalent HPV vaccine includes HPV 6, 11, 16, and 18, which are the most common low-risk subtypes (6, 11) and high-risk subtypes (16, 18). There is also the 9-valent vaccine which protects against HPV 6, 11, 16, 18, 31, 33, 45, 52, and 58. This vaccine is currently recommended for children aged 11–12 per the Center for Disease Control. The vaccine is usually given until the age of 26. A discussion about the risks and benefits should be had for adults aged 27–45 with the possibility that they have already been exposed to these strains of HPV [41]. A study by Palefsky et al., randomly assigned healthy men who have sex with men to either receiving the quadrivalent HPV vaccine or placebo. They identified a reduced rate of grade 2 and 3 AIN in the vaccine group with no significant vaccine related side effects [42]. Given the success of reduction in the precursor lesions with vaccination it will be interesting to see the effects on the incidence of new anal cancer cases over the next decade.

## 7. Conclusions

There have been significant changes in the management of anal cancer over the last 30 years due to the findings in the described clinical trials. The overall goal of each trial was to improve the morbidity and mortality associated with anal cancer treatment—either by altering first-line treatment or mitigating the side effect profile. The current regimen of concurrent chemoradiation with 5-FU and mitomycin with intensity modulated radiation therapy has been well published and substantiated. There has been significant change and improvement in management from the days of mandatory abdominoperineal resections to our current regimen with excitement for the ongoing studies, including DECREASE and Personalizing Radiotherapy Dose in Anal cancer. Even though there have been countless trials studying this rare malignancy, there is still work to be done, especially with inclusion of HIV positive patients. Additionally, as more focus turns to immunotherapy and individualized treatment plans based on tumor biology, it will be crucial to measure its impact on anal cancer.

## Figures and Tables

**Figure 1 cancers-17-01289-f001:**
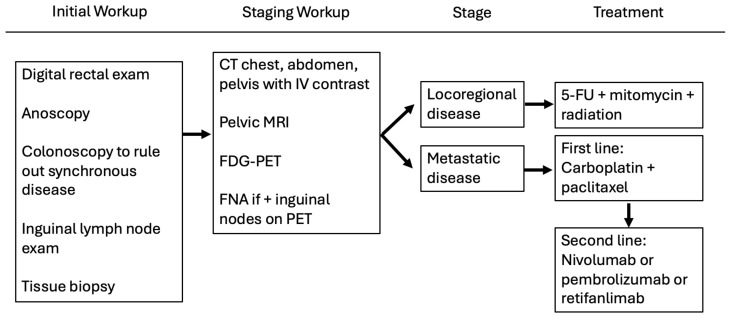
Initial work-up and treatment algorithm for anal cancer. The figure consists of the initial work-up and staging for patients suspected to have squamous cell carcinoma. The treatment algorithm changes based on the stage of disease with patients with locoregional disease undergoing Nigro protocol with 5-fluorouracil (5-FU), mitomycin and concurrent radiation. Patients with metastatic disease are initially treated with systemic chemotherapy.

**Figure 2 cancers-17-01289-f002:**
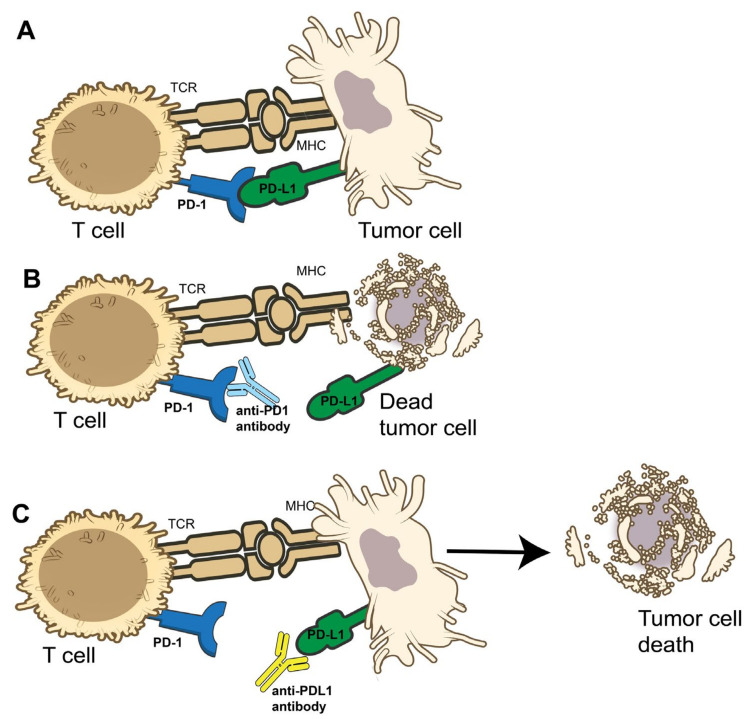
Mechanism of PD-1 and PD-L1. (**A**) The interaction between PD-1 on T cells and PD-L1 on tumor cells. When PD-1 and PD-L1 are bound to each other, the T cells are inhibited and do not attack the tumor cells. (**B**) The T cell’s PD-1 receptor is bound by an anti-PD1 antibody (i.e., pembrolizumab or nivolumab). This interaction with the drug contributes to the T cell attacking the tumor cell, causing cell death. (**C**) The tumor cell’s PD-L1 is bound by an anti-PD-L1 antibody (i.e., avelumab), preventing the interaction with PD-1. The T cell is not inhibited in this interaction and causes subsequent tumor cell death. Images created with the use of NIH BIOART source [27].

**Table 1 cancers-17-01289-t001:** American Joint Committee on Cancer Version 9 staging for anal cancer (2023).

Stage	T	N	M
I	T1	N0	M0
IIA	T2	N0	M0
IIB	T1-T2	N1	M0
IIIA	T3	N0–N1	M0
IIIB	T4	N0	M0
IIIC	T4	N1	M0
IV	Any T	Any N	M1

**Table 2 cancers-17-01289-t002:** Clinical trials for management and treatment of anal cancer.

Clinical Trial Name	Stage of Anal Cancer	Treatment Regimens	Results
ACCORD 03	Non-metastatic anal cancer	Induction chemotherapy with concurrent chemoradiation and standard radiation boost (Arm A) OR induction chemotherapy with concurrent chemoradiation and high dose boost (Arm B) OR concurrent chemoradiation with standard boost (Arm C) OR concurrent chemoradiation with high dose boost (Arm D) (All chemotherapy regimens: 5-FU and cisplatin)	Colostomy free survival: Arm A: 69.6% Arm B: 82.4% Arm C: 77.1% Arm D: 72.7%
ACT I	Stage II-III, non, metastatic	Radiotherapy alone OR concurrent chemoradiation with 5-FU and mitomycin	Local failure rate in the concurrent chemoradiation group of 39% compared to the radiation alone group of 61% at 3 years
ACT II	I-III, non-metastatic disease	Radiation in combination with Cisplatin +5-FU OR mitomycin + 5-FU with or without maintenance chemotherapy	Complete response post treatment with mitomycin+ 5-FU: 90.5% vs. cisplatin+ 5-FU: 89.6%
CARACAS	IV or non-metastatic and failed a previous line of therapy	Avelumab monotherapy OR cetuximab + avelumab	Overall response rate: avelumab only: 10% vs. cetuximab + avelumab: 17%
INTERAACT	IV	Cisplatin +5-FU OR carboplatin + paclitaxel	Objective response rate: cisplatin + 5-FU 57% and carboplatin + paclitaxel 59%
KEYNOTE-28	Locally advanced or metastatic, PD-L1 positive tumors	Pembrolizumab	Overall response rate: 17% Disease control rate: 58%
KEYNOTE-158	Locally advanced or metastatic SCCA who had a previous failure or intolerance to standard therapy	Pembrolizumab	Objective response rate of 11%
POD1UM-202	Advanced or metastatic, previously treated	Retifanlimab	Overall response rate in 13.8% and stable disease in 35.1%
RTOG 9811	Stage II-III (T2-4, N0-3, M0)	Radiation in combination with Cisplatin + 5-FU OR mitomycin + 5-FU	5-year disease free survival: mitomycin group 60% and cisplatin group 54%

**Table 3 cancers-17-01289-t003:** NCCN surveillance recommendation.

Nigro Protocol Algorithm							
Next steps	Physical exam, DRE at 8–12 weeks						
Outcomes	**CCR**	**Persistent disease**				**Progressive disease**	
Next steps	Surveillance	Exam and DRE in 4 weeks				Biopsy and restage	
Outcomes		CCR *	Regression or no progression		Progressive Disease ^†^	Local recurrence or inguinal disease	Distant Metastases
Next Steps			Exam Q3 months			APR or inguinal lymph node dissection Consider immunotherapy instead of APR.	Systemic therapy
Outcomes			CCR *	Persistent or Progressive Disease ^†^			
**Surveillance Protocol ** After APR, local excision for T1 disease with adequate margins, Nigro protocol with CCR							
**Parameter**		**Frequency**					
DRE, inguinal lymph node exam		Every 3–6 months for 5 years					
Anoscopy		Every 6–12 months for 3 years					
CT C/A/P ^‡^ with contrast		Annually for 3 years					

Abbreviations: DRE, digital rectal exam; APR, abdominoperineal resection; CT C/A/P, computed tomography chest, abdomen, pelvis; s/p, status post; CCR, complete clinical response. * Proceed to CCR column. ^†^ Proceed to Progressive disease column. ^‡^ Yearly CT chest with contrast and abdominal/pelvic MRI with contrast is an acceptable alternative.

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
