# Peer review of "Advances in the Management, Treatment, and Surveillance of Anal Squamous Cell Cancer"

_cancers, 2025, doi:10.3390/cancers17081289_

Round 1

Reviewer 1 Report

Comments and Suggestions for Authors

The manuscript is a concise documentation of the currently available data on Anal canal Cancer management and the evidence that guides the management approach. The authors have tried to incorporate most of the available data in the review.  A couple of points worth adding include:

  1. The role de-escalation with regards to Mitomycin- 2 vs 1 dose, with reference to the ACT II study and EORTC study.
  2. The role of chemoRT in the setting of limited metastatic disease

Author Response

Comment 1: The role de-escalation with regards to Mitomycin- 2 vs 1 dose, with reference to the ACT II study and EORTC study.

Thank you for pointing out the importance of de-escalation. On page 8, the ACT II study is discussed. This trial demonstrated that the use of maintenance chemotherapy was not beneficial and supports the idea of de-escalation for treatment. We have also discussed ACT 3 and 4 studies and the DECREASE study in future directions which would test multiple types of de-escalation therapy. Given the EORTC study by Bartelink et al was similar to ACT 1 we did not feel it provided much to the content of this paper in relation to de-escalation. If there is a specific EORTC study that the reviewer feels would be critical to include in this review we would be happy to make the appropriate additions.

Comment 2: The role of chemoRT in the setting of limited metastatic disease

Thank you for this suggestion. It would have been interesting to explore this aspect. Unfortunately, there are no prospective randomized control trials or rigorous data that review the use of chemoradiation in limited metastatic disease which is why this information was excluded. We did include one study that utilized the National Cancer Database on page 10 but feel no definitive conclusion can be made about the role of chemoRT with metastatic disease.

Reviewer 2 Report

Comments and Suggestions for Authors

This manuscript offers a comprehensive and well-organized review of the advances in the management, treatment, and surveillance of anal squamous cell cancer. The authors provide an in-depth synthesis of the literature, covering both historical and current therapeutic strategies, from the seminal Nigro protocol to contemporary approaches that incorporate immunotherapy. The detailed summaries of pivotal clinical trials—including RTOG 98-11, ACT I, ACCORD 3, ACT II, and emerging immunotherapy studies—help elucidate the evolution of treatment paradigms and their impact on patient outcomes.

The structure of the manuscript is commendable, with clear sections addressing epidemiology, diagnostic criteria, treatment regimens, and follow-up strategies. The inclusion of well-designed tables and figures enhances the clarity of complex data, making it accessible to a broad clinical audience. The language is precise and the arguments are logically developed, which underscores the rigor of the review.

Despite its strengths, there are areas that could benefit from further refinement. First, while the review effectively outlines clinical trials, it would be valuable to include a more critical evaluation of the methodological limitations inherent in these studies, particularly concerning patient selection criteria (e.g., the underrepresentation of HIV-positive individuals) and variations in treatment protocols. Second, the discussion on emerging immunotherapeutic strategies could be expanded to address challenges such as biomarker variability and the need for personalized treatment plans. Finally, a more detailed exploration of long-term surveillance strategies and the integration of novel imaging modalities would provide a holistic perspective on patient management.

Overall, the manuscript represents a significant contribution to the field of anal cancer research. With minor revisions to address these suggestions, the paper will serve as an excellent resource for clinicians and researchers seeking to understand and advance the treatment of anal squamous cell cancer.

Author Response

Comment 1: First, while the review effectively outlines clinical trials, it would be valuable to include a more critical evaluation of the methodological limitations inherent in these studies, particularly concerning patient selection criteria (e.g., the underrepresentation of HIV-positive individuals) and variations in treatment protocols.

Thank you for this comment. We agree and have added a couple paragraphs comparing these studies on page 9.

Comment 2: Second, the discussion on emerging immunotherapeutic strategies could be expanded to address challenges such as biomarker variability and the need for personalized treatment plans.

Thank you for this suggestion, we have incorporated the use of 2 of the biomarkers, ctDNA and tumor infiltrating lymphocytes into the review on page 14-15. We feel a full analysis of all biomarkers is beyond the scope of this review.

Comment 3: Finally, a more detailed exploration of long-term surveillance strategies and the integration of novel imaging modalities would provide a holistic perspective on patient management.

Thank you for this comment and appreciate your feedback. We have added some additional information to the text about more long-term surveillance for anal cancer patients on page 14. We feel Table 3 is a comprehensive summary of the surveillance schedule. After a comprehensive literature search, we were unable to identify novel imaging modalities. We would be happy to include this information if there is a certain imaging modality that is suggested.

Reviewer 3 Report

Comments and Suggestions for Authors

This review covers key clinical trials on treating and monitoring precancerous anal lesions and anal squamous cell cancer, highlighting changes in radiation, chemotherapy, and immunotherapy over recent decades. The review is well structured and provides a fairly comprehensive overview of the current treatment of SCCA.

My comments and suggestions:

1. The review barely described the surgical role in treating SCCA.

2. The authors limit their consideration of immunotherapy to anti-PD1 therapy. Additional immunotherapy targets like T-lymphocyte antigen 4 (CTLA4) (ipilimumab), LAG3 (relatlimab), and CD38 (daratumumab) should also be considered. The same goes for adaptive cell therapy and viral therapy. Overall, this review lacks a scope for the immunotherapeutic future.

3. The MS is somewhat overloaded with abbreviations, which makes it difficult to read.

Author Response

Comment 1: The review barely described the surgical role in treating SCCA.

Thank you for pointing this out. Initially, the surgical role in treating SCCA was excluded as we wanted to focus on high quality clinical trials that have shaped the management of anal cancer and were unable to identify any trials that included surgical management. We have amended section 1.3: “Nigro protocol and current NCCN guidelines” to “Anal Cancer Treatment and NCCN guidelines” to better reflect the variety of treatment options. On page 3, we have included a discussion on local excision and abdominoperineal resection.

We hope in the near future when the ACT 3 study, which is mentioned in the future directions, is published it will provide valuable data to the field about the role of local excision.

Comment 2: The authors limit their consideration of immunotherapy to anti-PD1 therapy. Additional immunotherapy targets like T-lymphocyte antigen 4 (CTLA4) (ipilimumab), LAG3 (relatlimab), and CD38 (daratumumab) should also be considered. The same goes for adaptive cell therapy and viral therapy. Overall, this review lacks a scope for the immunotherapeutic future.

We appreciate this suggestion and had initially excluded many other trials as we felt a comprehensive immunotherapy review was beyond the scope of this manuscript. We acknowledge that there are many studies that utilize immunotherapy and have included the CARACAS study in the manuscript. We also included some data from the POD1UM-303 to supplement the data from POD1UM-202 and NCI 9673 Part B that assessed the utility of ipilimumab in combination with nivolumab. Other immunotherapy and bioengineered therapies have also been added to recognize the remarkable work being done in this field.  This information is found on pages 12-13.

Comment 3: The MS is somewhat overloaded with abbreviations, which makes it difficult to read.

Thank you for pointing this out. We agree with this comment and have removed multiple abbreviations throughout the manuscript, including LAST, AJCC, PLATO, CTV, DP-IMRT, CCR, and. DRE. Throughout the manuscript the changes have been made and are highlighted.